# Microencapsulation of Betalains Extracted from Garambullo (*Myrtillocactus geometrizans*) to Produce Active Chitosan–Polyvinyl Alcohol Films with Delayed Release of Bioactive Compounds

**DOI:** 10.3390/antiox13091031

**Published:** 2024-08-25

**Authors:** Daniela Gómez-Espinoza, J. A. Gonzalez-Calderon, Ricardo Rivera-Vázquez, César Leobardo Aguirre-Mancilla, Enrique Delgado-Alvarado, Agustín L. Herrera-May, Ma. Cristina Irma Pérez-Pérez

**Affiliations:** 1Departamento de Ingeniería Bioquímica, TecNM en Celaya, Celaya 38010, Mexico; d2303031@itcelaya.edu.mx; 2Cátedras CONACYT-Instituto de Física, Universidad Autónoma de San Luis Potosí, San Luis Potosí 78290, Mexico; amir@ifisica.uaslp.mx; 3Instituto Nacional de Investigaciones Forestales, Agrícolas y Pecuarias, Campo Experimental Bajío, Celaya 38110, Mexico; rivera.ricardo@inifap.gob.mx; 4TecNM en Roque, Celaya 38110, Mexico; cesar.am@roque.tecnm.mx; 5Micro and Nanotechnology Research Center, Universidad Veracruzana, Boca del Rio 94294, Mexico; endelgado@uv.mx (E.D.-A.); leherrera@uv.mx (A.L.H.-M.); 6Facultad de Ingeniería de la Construcción y el Hábitat, Universidad Veracruzana, Boca del Rio 94294, Mexico

**Keywords:** biopackaging, betalains, chitosan–polyvinyl alcohol (PVOH) films, garambullo, microencapsulation, preservation, tomato coatings

## Abstract

Garambullo is a plant with little industrial application. However, garambullo contains betalains, photosensitive phytochemical compounds, which through microencapsulation can be used in chitosan–polyvinyl alcohol (PVOH) films for application in tomato coatings. These biopackages were characterized by physical tests, water vapor permeability, puncture tests, extension, color, differential scanning calorimetry (DCS), Fourier transform infrared (FTIR) spectroscopy, and antioxidant and antimicrobial activity analyses. The influence of the biopackages on the tomato coatings was measured using parameters such as minimum weight loss close to 2% at day 9, pH of 4.6, Brix of 5.5, titratable acidity of 1 g acid/100 mL sample, and shelf life of up to 18 days. The biopackages containing betalain microcapsules had a water vapor permeability of 2 × 10^−14^ g/h·m·Pa and an elongation of 5 ± 0.5%, indicating that the package did not stretch. The deformation at the breaking point for the package without and with microcapsules was 0.569 and 1.620, respectively. With respect to color, adding white microcapsules and betalains can cause the material to darken, resulting in a yellowish color. Furthermore, the phenolic content was greater for the biopackages with betalains, while there was no significant difference in the antioxidant activity since the active compounds were not released. According to the in vitro results, the inhibition of *B. cinerea* was achieved on the eighth day when the active compounds were released from the microcapsules. The tomato with betalains lost 2% of its weight, and *B. cinerea* was inhibited, extending its shelf life to 18 days. The proposed biopackages have good properties as biopolymers and inhibit the presence of *B. cinerea*.

## 1. Introduction

Garambullo (*Myrtillocactus geometrizans*) is a cactus distributed in central Mexico, which can be considered as functional food due to its content of bioactive compounds that are hypoglycemic and hypocholesterolemic [1,2]. Among these phytochemical compounds are betalains, which are used for pigmentation. The betalains are natural colorants derived from indoles [3] and composed of a central nitrogenous structure known as betalamic acid [4-(2-oxoethylidene)-1,2,3,4-tetrahydropyridin-2,6 dicarboxylic acid]. Betalains are classified according to their composition into betaxanthins and betacyanins, representing yellow and purple-red colors, respectively [4]. In addition, several researchers [5] have reported the betalains antioxidant properties because the phenolic groups within betalains are electron donors, obtaining a high capacity to scavenge free radicals. Unfortunately, these compounds are unstable with light, temperature, pH, enzymatic activity, and the presence or absence of oxygen and metals. Therefore, an optimal process is required to preserve the properties of the betalains [6]. For instance, microencapsulation by spray drying can be used to conserve the properties of the betalains. Microencapsulated materials are considered miniature packaging, which can be sealed, and their content can be released at controlled rates under specific conditions [7]. On the other hand, drying involves rapidly evaporating moisture and maintaining a low particle temperature. Furthermore, microcapsule formation consists of homogenizing the core materials and encapsulating materials, generating an emulsion that is atomized in the drying chamber [8]. Food packaging using plastic materials can cause a global pollution problem. Biopackaging can be an alternative solution to reduce this pollution problem. Biopackaging involves packaging that protects the food from the environment and other physical and chemical factors, ensuring the quality and safety of food products and allowing their transport and prolonged storage [9].

The biopackages can be fabricated from edible or biodegradable polymers such as chitosan and polyvinyl alcohol (PVOH), which could reduce plastic waste and environmental impact [10]. Chitosan is the main derivative of chitin, which is an amino-polysaccharide composed mainly of repetitive units of 2-amino-2-deoxy-D-glucopyranose [11]. Barik et al. [12] reported that chitosan-based films can extend the shelf life of foods, including good properties such as color, gloss, resistance to UV light, resistance to water, water permeability, resistance to traction, elongation, and breakage [12]. On the other hand, polyvinyl alcohol is a synthetic and biodegradable polymer containing hydroxyl groups that promote its interaction with proteins, forming a structurally stable polymeric network with a high-water absorption capacity and water vapor permeability [13].

Antimicrobial activity can reduce the presence of microorganisms, such as bacteria and fungi [14]. For the betalains, the antimicrobial activity is due to polyphenolic extracts. For instance, a phenolic content of 5.26 ± 0.28 mg GEA/g^−1^ was obtained for the garambullo, a stage of consumption maturity with dark purple coloration [15], associating this content with its antimicrobial activity. *Botrytis cinerea* is a phytopathogenic, necrotrophic, saprophytic, and parasitic-pathogenic fungus that is known to attack more than 200 species of plants, especially horticultural and fruit plants, and numerous wild plants, mainly dicotyledonous [16]. This disease can infect tomato plants at any vegetative stage, including postharvest [17].

In this work, we propose an effective method to preserve the main components in garambullo juice using a microencapsulation process. We present the influence of microencapsulated betalains based on garambullo in the fabrication of chitosan–PVOH films and their application in tomato coating. The performance of these biopackages was characterized using physical tests, water vapor permeability, puncture tests, extension, color, differential scanning calorimetry (DCS), Fourier transform infrared (FTIR) spectroscopy, and antioxidant and antimicrobial activity analyses. This process can enable the long-time delivery throughout biopolymeric films to generate a novel biocomposite with application in the food industry.

## 2. Materials and Methods

### 2.1. Chemicals and Reagents

Mucilage (Aloe vera), glycerol (J.T. Baker Lot: M16C52, CDMX, Mexico), globe maltodextrin 10 (Ingredion code: 10520015, Guadalajara, Mexico), silicon dioxide (Agroin, Celaya, Mexico), starch (Ingredion Batch: FKI6714, Guadalajara, Mexico), garambullo juice, Tween 20 (azumex universal code: 07503031232370, CDMX, Mexico), Tween 80 (HYCEL lote: 263534, CDMX, Mexico), chitosan medium molecular weight Lot: STBF4197V Sigma-Aldrich and Poly (vinyl alcohol) Lot: MKBH1410V Sigma-Aldrich, St. Louis, MO, USA.

### 2.2. Microcapsules Production and Physico-Chemical Characterization

Microcapsules were formed by preparing an emulsion with a maximum content of 30% soluble solids, which were accepted in the spray-drying process, composed by 11.03% garambullo juice, 0.95% *Aloe vera* mucilage, 9.48% starch, 0.1% Tween 20, 7.11% maltodextrin and 1.42% SiO_2_ (MBT), and the control microcapsules were prepared by the same formulation without garambullo juice (MB) using a spray-drying temperature of 160 °C.

#### 2.2.1. Determination of the Antioxidant Capacity of Microcapsules and Films

##### Quantification of Betalains in Microcapsules

Quantification of betalains in microcapsules was performed according to the process reported in the literature [18,19]; 10 mg of microcapsules was weighed and placed in a flask and dispersed with 40 mL of distilled water under vortex agitation for 20 min, then centrifuged at 490 rpm for 5 min and filtered, then centrifuged at 490 rpm for 5 min and filtered on Whatman paper No. 4 to obtain the absorbance reading for betacyanin at 538 nm and for betaxanthin and betaxanthins at 472 nm; the determinations were performed in triplicate. The following equation was used:(mg of pigmento)/(100 g of sample (b.h.)) = (*A*)(*Fd*)(*PM*)/*E*(1)
where *A* is the absorbance at 476 nm for betaxanthins and 538 for betacyanins and *Fd* is the dilution factor that for this case is 0.25, *PM* is the molecular weight of the pigment, betacyanins (550.5 g/mol) and betaxanthins (339.3 g/mol), *E* is the molar extinction coefficient, betacyanins (1120 L/mol·cm) and betaxanthins (750 L/mol·cm).

##### Phenolic Content

The phenolic content was determined according to the previous method reported in the literature [20] with some modifications; 25 μL of the sample, 25 μL of Folin–Ciocalteau reagent, and 25 μL of sodium carbonate were added to a microplate. The mixture was incubated for 30 min at 40 °C, 200 μL of distilled water was added, and the absorbance was read at 750 nm. The calibration curve was generated with a gallic acid (GA) solution. The results are expressed in mg gallic acid equivalents per gram of dry weight of the sample (mg GAE/g sample). The following equation was used:(GAE mg)/(100 g) = ((mg/mL)(v extraction))/(g sample)(2)

##### ABTS ((Ácido 2,2′-Azino-bis-(3-etillbenzotiazolin-6-sulfonic)) Antioxidant Activity

The ABTS method was evaluated according to the procedure proposed by [21]. The reading was performed in a 96-well plate, in which 20 µL of Trolox + 230 µL of the working solution was added for the sample measurement. The mixture was allowed to stand for 6 min, after which the absorbance was read at 734 nm. The samples were read at 4, 10, 30, 60, and 90 min. The calibration curve was generated with a Trolox 800 µM solution. The results are expressed in mg Trolox equivalents per gram of dry weight of the sample (mg Trolox/g sample). The following equation was used:(µM Eq. of Trolox)/g = ((mg/mL)(v extraction))/(g sample)(3)

##### DPPH (2,2-Difenil-1-picrilhidrazil) Determination

The DPPH method was assessed according to the procedure proposed by [22]. The reading was performed in a 96-well plate, where 20 µL of extract and 280 µL of radical were added for sample measurement. In addition, 20 µL of absolute methanol and 280 µL of radical were used as blanks; after 30 min, the mixture was allowed to stand in the absence of light, and the reading was obtained at 515 nm. The calibration curve was generated with a Trolox 800 µM solution. The results are expressed in mg Trolox equivalents per gram of dry weight of the sample (mg Trolox/g sample); Equation (3) was used.

##### FRAP (Ferric Reducing Antioxidant Power) Determination

The FRAP method was evaluated according to the procedure proposed by [23]. In a 96-well microplate, 20 µL of extract and 280 µL of radical were added, and the mixture was allowed to stand for 30 min in the absence of light; the absorbance was read at 593 nm. The calibration curve was generated with a Trolox 800 µM solution. The results are expressed as mg Trolox equivalents per gram of dry weight of the sample (mg Trolox/g sample); Equation (3) was used.

### 2.3. Scanning Electron Microscopy (SEM) Analysis

The surface morphology of the films was examined by high-resolution scanning electron microscopy (SEM) (XL30-SFEG, Philips/FEI, Hillsboro, OR, USA). Before examination, the microcapsules were coated with a thin layer of gold and inspected at an acceleration voltage of 10 kV.

### 2.4. Bio-Polymeric Film Fabrication

The film-forming chitosan solution was prepared according to the procedure reported by [24]. The resulting solutions were prepared as follows: 7.5 g of chitosan (CS) dissolved in a 1% (*w*/*v*) acetic acid solution (500 mL) was constantly stirred and heated for 30 min at 50 °C to improve the solubility of chitosan. A 4% (*w*/*v*) polyvinyl alcohol (PVOH) solution was prepared in distilled water at 80 °C. Chitosan and polyvinyl alcohol were mixed at a CS 70%-PVOH 30% ratio at room temperature under magnetic stirring for 30 min. Then, 2% glycerol in the CS base was added and stirred for 10 min. Finally, the betalain microcapsules were incorporated at a 1:1 ratio by weight on a CS basis and kept under magnetic stirring for 30 min. Each forming solution was placed in Petri dishes (5 and 10 cm), dried under laboratory conditions (23–26 °C) for approximately 48 h, and peeled off the plates for further analysis (Figure 1). The negative control was a mixture, and the positive control was a mixture of CS 70%-PVOH 30% with control microcapsules (microcapsule formation without betalains).

### 2.5. Physical Characterization of Films

The weight of the films was measured to determine the density and surface density of the material using an analytical balance (Sartorius-Werke AG, Gottingen, Germany) with an accuracy of 0.0001. The surface density was calculated as the ratio of the weight of the sample with respect to the area of 1.26 × 10^−3^ m^2^. The density of the material was obtained from the relationship between the surface density and the thickness of the material. The water vapor permeability of the films was determined according to the method presented in the literature [25]. Permeability cells formed by a glass container were used. Furthermore, a perforated screw cap that allowed an exposure area of 9.079 × 10^−4^ m^2^ and two Teflon gaskets ensured an airtight seal, between which the film was placed. The permeability cell containing 15 mL of distilled water (RH_1_ = 100%) was placed in a vacuum desiccator with silica gel as a desiccant that maintained an external relative humidity (RH_2_) of 34%. The evaluated films were cut with a diameter of 4 cm, and the thickness was determined in 5 points measured randomly using a micrometer (Foil Dial Thickness Gauge F 1101/30, KAFER GmbH of Villingen-Schwenningen, Germany). The jars were weighed at regular time intervals. The kinetics of cell weight variation as a function of time until reaching equilibrium were recorded.

### 2.6. Mechanical Properties

It was determined on a texture analyzer TA-XT2i coupled to Texture Expert Exceed version 2.63 software. To determine the puncture stress, the samples were cut in a circular shape with a diameter of 4 cm and placed in a cell. These samples were well secured in the texturometer perpendicular to the film’s surface, and the P/2 cylindrical probe of 2 mm diameter of stainless steel was used. The speed of the head was 2 mm/s, a distance of 40 mm, and an activation force of 5 g. Calculating the maximum stress at the film puncture rupture according to the following equation:(4)$$\frac{\Delta l}{lo}=\frac{\sqrt{D^{2}+{lo}^{2\;}}-lo}{lo}$$
where *D* is the probe displacement and *lo* is the initial length of the film

The tensile behavior was measured using the following procedure: the samples were trimmed, leaving a length of 8 cm between the edges, and the tape was placed to hold the probe. For the elongation process in the texturometer, the tensile grips probe was used with a uniaxial velocity of the head of 0.05 mm/s, a distance of 30 mm, and a force of 15 g. Thus, tensile strength (MPa), Young’s modulus (MPa), and elongation (%) can be measured [26].

### 2.7. Color Determination

Color determination was carried out using the Chroma Meter CR-400 (konica minolta Morristown, NJ, USA), and the color coordinates were evaluated according to the scale proposed by the CIE L a b chromaticity and hue degrees.

### 2.8. Fourier Transform Infrared Spectroscopy (FTIR)

The sample was placed in a cell closed and isolated from the environment. In addition, infrared patterns were recorded on ThermoFisher Scientific Nicolet FTIR 6700 spectrometer, Markham, ON, Canada with a DRIFT Spectra-Tech collector equipped with a high-temperature heating cell. Scanning from 400 to 4000 cm^−1^.

### 2.9. Differential Scanning Calorimetry (DSC) Analysis

DSC was used to carry out isothermal crystallization for the iPP and its composites. The equipment used was a TA Instruments DSC (New Castle, DE, USA), model Q2000, with temperature ranges from −70 to 400 °C. Approximately 10 mg of sample was subjected to temperature cycles, performing a heating ramp from 30 °C/min to 270 °C, isotherm for 5 min with a ramp of 10 °C/min under nitrogen atmosphere.

### 2.10. Antimicrobial Activity

*B. cinerea* was obtained from the “Instituto Nacional de Investigaciones Forestales Agrícolas y Pecuarias” (INIFAP) provided by Dr. Luis Antonio Mariscal Amaro, extracted from strawberries.

They were grown for 6 days at 19 °C on potato-dextrose agar plates (PDA, MCD LAB, San Jacinto Amilpas, Mexico) acidified with 10% p/v tartaric acid. They were used to recover fungal spores by pouring 25 mL of sterile distilled water over the surface of the agar plate, followed by gentle scraping with a sterile rake to remove the maximum number of spores. Spore suspensions were transferred to tubes containing 9 mL of sterile PDA using the puncture methodology. They were cultured for 6 days at 19 °C. They were subsequently used to recover fungal spores by pouring 10 mL of sterile distilled water with 0.01% Tween over the agar surface, followed by gentle scraping using a sterile rake to remove the maximum number of spores. The number of spores in the suspension was determined using a hemocytometer and an optical microscope, which can be expressed as the number of spores per milliliter (spores/mL). Thus, suspensions were serially diluted to approximately 1 × 10^6^ spores/mL [27].

#### Disc Contact Assay

The inverted lid technique [28] was used; 100 μL of spore suspension (1 × 10^6^ spores/mL) was placed in the center of the PDA plate and dried in a laminar flow hood under aseptic conditions. The PDA plate was dried in a laminar flow hood under aseptic conditions at room temperature for 30 min, then film discs corresponding to antimicrobial film to cover the plate with growth agar were measured every 24 h for 8 days of incubation at 20 °C. Each assay was performed in triplicate.

### 2.11. Application on Tomato

The raw material was used in ball tomatoes, which were selected by size, maturity, absence of physical damage, and fungal infections. Next, the tomato pedicle was cut and sanitized using a 0.35% *p*/*v* colloidal silver solution (Microdyn, CDMX, Mexico).

### 2.12. Evaluation of the Coating of CS-PVOH and Microcapsules of Betalains on Tomato

Preliminary tests were carried out on tomatoes to evaluate the behavior of the coating on tomato physicochemical parameters.

The filmogenic suspension was prepared according to Section 2.2. The tomatoes were coated with direct applications of the filmogenic suspension, dipping and adding them with a brush on the stalk side. The coating was dried with Lakewood fans (250 V, 50/60 Hz). After coating, they were kept refrigerated at a temperature range from 4 ± 1 °C for the period necessary for each test.

### 2.13. Physico-Chemical Properties of Coated Tomatoes

#### 2.13.1. Weight Loss (WL)

The fruit weight of each replicate was recorded on the day of treatment and subsequent days of sampling.

#### 2.13.2. pH Determination

The pH was determined according to [29], which was measured in the tomato juice of each sample using a pH-meter (Oakton, pH/Mv/°C meter, Singapore)

#### 2.13.3. Soluble Solids (SS)

The determination of soluble solids was carried out on juice samples of tomato for each treatment at 25 °C in triplicate, in a refractometer (Pocket pal-1, Atago (0–53%, Shanghai, China)); the results are expressed as Brix index (°Brix).

#### 2.13.4. Titratable Acidity (TA)

The titratable acidity (TA) was obtained according to [29], based on titration of approximately 3 g of fruit puree. AOAC (1996), based on the titration of approximately 3 g of fruit puree with 0.1 N NaOH solution, pre-titrated with 3 drops of 1% phenolphthalein and using a potentiometer to a pH value of 8.2 ± 0.2. The result is expressed as a percentage of citric acid and calculated according to according to the equation:(5)% acidity=V·N·miliequivalent
where % acidity is the g of acid/100 mL of sample, V is the volume (mL) of NaOH used for titration, N is the normality of NaOH, and milliequivalent is the quantity of the predominant acid in the food equivalent to 0.001 mL of 0.1 N NaOH 0.1 N, which in the case of citric acid is equivalent to 6.4 g

### 2.14. Shelf Life Determination

Tomatoes dip-coated with the different biopackages were subjected to refrigeration (4 °C) for 18 days, visually assessing the growth of *B. cinerea* on the surface of the samples.

### 2.15. Statistical Analysis

The results are expressed as the mean ± standard deviation of the experiments with three replicates. The statistical evaluation was carried out by analysis of variance (ANOVA), comparing means by Tukey’s method with a significance level of *p* < 0.05, using STATGRAPHICS centurion XVI.I software V. 16 no.

## 3. Results and Discussion

The microcapsules with betalains from garambullo were characterized by color, total betalains, betaxanthins, betacyanins, water activity, total phenols, and antioxidant activity (DPPH, ABTS and FRAP). These experimental results are summarized in Table 1, Table 2 and Table 3.

### 3.1. Microparticle Morphology (SEM)

The morphology of the control (white) and betalain microcapsules were observed in SEM images, as shown in Figure 2.

Figure 2 shows the microcapsules containing betalains from the garambullo treatment and the control at 160 °C. These microcapsules depict variable shapes, such as spherical-shaped capsules with smooth surfaces and capsules with irregular surfaces (sizes between 5 and 10 μm). Generally, the formation of dents on the capsule surface is caused by the shrinkage of the particles during drying. It is due to the drastic loss of moisture, followed by cooling [30]. Ref. [31] investigated that the spherical morphology is due to maltodextrin in the formulations. Low-molecular-weight sugars (e.g., maltodextrin) can act as plasticizers that reduce the number of polymer chain contacts, reducing the rigidity of the three-dimensional layered structure. In addition, Fernández-Repetto et al. [32] reported that higher solids content caused more roundness in the capsule shape. It is due to the combination of the encapsulating agents and the encapsulated material, which is shown in microcapsules with garambullo juice inside.

### 3.2. Mechanical Characterization of Biopackages

Table 4 depicts the parameters of a group of samples, including the negative control (C-1) and the positive control (C+). The parameters of both control samples do not have significant difference. However, the samples with microcapsules of betalains registered a significant difference.

### 3.3. Water Vapor Permeability

Figure 3 depicts the values of water vapor permeability of the proposed biopackages, according to [33]. This parameter must decrease to avoid the moisture transfer between food and the environment, which can increase the shelf life of the food. The results of water vapor permeability of our biopackages are lower than those reported by [33,34], which were based on chitosan and betalains.

### 3.4. Puncture Stress and Strain at Extension

Table 5 shows the characteristics of the puncture stress and tensile strain. The samples containing betalain microcapsules showed no significant difference in Young’s modulus and tensile strength. However, there was a significant difference in the % elongation, and the values were very close, which indicates that the packaging did not stretch, as there was cohesion between the chains. According to the literature [35], the same behavior was observed for samples of chitosan–gelatin films supplemented with tyrosol and encapsulated ferulic acid, while the tensile strength decreased without microcapsules. On the other hand, Kurek et al. [36] studied a functional packaging of chitosan and pectin with encapsulated *Opuntia-ficus indica* residues, reporting that variations in the results of the biopackages can be due to the nature and source of the film-forming polymer (impurities, degree of deacetylation of chitosan, esterification of pectin, and so on). These parameters vary according to the study, the type and content of the plasticizer, and the intermolecular forces; at the time of puncture, there is a difference as it is in a direction perpendicular to the manufactured film, in which the microcapsules are all over the packaging.

### 3.5. Determination of Color in Biopackaging

Table 6 indicates the results of evaluating the color parameters in the biopackaging films. The statistical analysis shows significant differences for the parameters measured. These differences indicate that the addition of both white and betalain microcapsules tends to darken, acquiring a yellowish color. This result agrees with the literature [37], in which the chitosan biopackage tends to have a yellowish tone because the microcapsules have betalains, divided into betacyanins and betaxanthins. It gives the yellow colorations; furthermore, the value for the parameter a* is positive and higher for those with microcapsules. It is due to the microcapsules with betalains, and the tone would be associated with the presence of betacyanins, which are associated with the red-violet colors [38], presenting a higher saturation for the samples with betalains.

### 3.6. Fourier Transform Infrared (FTIR) Spectroscopy

Figure 4 depicts the FTIR spectrum of the biopackaging films, in which the characteristic absorption bands of the OH, C-O, C-H, CH_2_, and C=O functional groups of the chitosan (CS) and PVOH are observed. The band at 3274 cm^−1^ corresponding to -OH is more pronounced for PVOH than for CS. CS has a pronounced band at 2915 corresponding to -C-H; while for PVOH, it is at 2940 cm^−1^. At 1745 cm^−1^, a band corresponding to (CH)-CH_2_ is observed for CS. At 1573 cm^−1^, the band in CS corresponding to OH-C-OH is visualized, while for PVOH, at 1409 cm^−1^, the band is of N-H. Also, for PVOH at 1020 cm^−1^, a peak corresponding to (C-O)-C-OH is illustrated and was identified at 1087 cm^−1^, -C-O-C in PVOH at 1328 cm^−1^, and in CS at 1382 cm^−1^, in the CS/PVOH mixture the OH band was more pronounced than for CS. The OH bands in the PVOH-CS system increase in size with a lower intensity, indicating a reduction in the crystallinity of PVOH to shift the position and intensity of the O-H peak, the hydrogen atoms of PVOH and the oxygen atom of CS establish hydrogen bonds with each other [39]. For (CH)-CH_2_, the peak is found at 1729 cm^−1^, which is less pronounced than for CS but higher than for that of PVOH. The (C-O)-C-OH is found at 1596 cm^−1^. In microcapsules, there was no change between microcapsules with betalains and those without betalains, according to [40]. Betalains show a band at 3400 cm^−1^ corresponding to O-H stretching, and at 2921 cm^−1^ associated with symmetric C-H stretching, at 1646 cm^−1^, 1411 cm^−1^, and 1253 cm^−1^ corresponding to asymmetric C=O, C-O, and C-O stretching of the acid, respectively. The band at 1023 cm^−1^ corresponds to the C-N stretching and the band at 754 cm^−1^ corresponds to the presence of the amide group. These changes in the bands could not be identified in the microcapsules with betalains, probably because the microencapsulation was properly performed, avoiding the betalains being exposed to be able to visualize them. When added to the packages in both cases, there was an extension of the (CH)-CH_2_ band and a decrease in the band corresponding to (C-O)-C-OH. There were no modifications concerning the incorporation of betalains when they were microencapsulated.

### 3.7. Determination DSC

Figure 5 shows the DSC thermograms of the batches of chitosan films, PVOH, the CS-PVOH blend, CS-PVOH-MB, CS-PVOH-MBT, MB, and MBT, giving a glass transition (Tg) of 80 °C for PVOH, as reported by the literature [41] which was 52.4 °C with a melting temperature (Tm) of 225 °C. This temperature is similar to that reported by [42] of 219.5 °C. For CS, the glass transition is 118 °C that is similar to that reported by [43] of 123 °C; also, this result agrees with this research since it does not show melting behavior in the lactic acid films, suggesting that the chitosan molecules were in an amorphous state, making that for the other biopackages did not show melting behavior. The value for the CS-PVOH mixture was 110 °C, for CS-PVOH-MB, it was 154 °C, and for CS-PVOH-MBT, it was 80 °C. The change in the glass transition of the latter two was modified due to the addition of the microcapsules. For the CS-PVOH-MB packaging, it is modified since the white microcapsules have a glass transition of 120 °C and have a degradation at 217 °C, while for MBT, these behaviors are not perceptible except for the degradation, which is similar to that of MB with 211 °C.

### 3.8. Determination of the Antioxidant Capacity of the Filmogenic Suspension

The phenolic content in the biopackages significantly increased in the samples with microcapsules. Both white ones, without active nuclei, showed a greater phenolic content than those with nuclei (see Figure 6). It may be due to the added concentration of betalains, which contain phenolic compounds [44]. Antioxidant activity is performed by these three determinations since DPPH shows antioxidant activity against lipophilic compounds, ABTS against hydrophilic compounds, and antioxidant activity by FRAP is against lipophilic and hydrophilic compounds. As a result, for the determinations DPPH and ABTS, there was no significant difference for the samples with control microcapsules and microcapsules with betalains, which was probably because the release of the microcapsules was due to dissolution or swelling. In the first one, the wall material dissolved due to the influence of external factors, and in the second one, it is due to the absorption of the surrounding liquid [8]. The core was released by placing the microcapsules in the packaging [45]. Before packaging formation, the microcapsules had values above 4000 GAE mg/100 g of the sample. After the packaging and dissolution of the same for the measurement, the degradation of the compounds took place when in contact with the environment and probably left microcapsules that would only be released by the fracture mechanism, in which the encapsulant breaks due to external or internal forces. When the microcapsules do not break, they do not release the compound [8], releasing the nucleus at the moment of placing the microcapsules in the packaging. The equation for the calibration curve is as follows: for total phenols, Y = 0.004x + 0.0663 and an R2 = 0.9833, for DPPH, Y = −0.0007 + 0.8446 and R2 = 0.9946, for ABTS, Y = −0.001x + 1.552 and R2 = 0.9519, and for FRAP, Y = 0.0018x + 1.3584 and R2 = 0.9805.

### 3.9. Antimicrobial Activity

On day 1, no growth of *B. cinerea* was observed for any of the samples (see Figure 7); on day 4, growth was inhibited, mainly for the sample, and on day 8, for C- and C+, the fungus completely covered the box. In the first days, the antimicrobial effect could be attributed to the antimicrobial activity of chitosan since it has only cationic (positive) charges. These charges are responsible for binding to anionic (negative) components in bacterial membranes, such as phosphoryl groups of phospholipid components, proteins, amino acids, and various lipopolysaccharides, through electrostatic attractions leading to the breakdown of cell membrane permeability [46]. However, the main active compound promoting the antimicrobial activity of this container is betalains, which are released on day 8 when a larger halo of inhibition occurs because the biopolymer decomposes, and microcapsules release the betalains, which have aromatic rings in their chemical structure that include elements in common with phenolic compounds [47]. These phytochemicals have been shown to have antimicrobial activity, as lipophilicity allows active phenols to penetrate biological membranes, while hydroxyl groups can act by uncoupling.

### 3.10. Coating Application on Tomato

#### 3.10.1. Physico-Chemical Properties of Tomato

The titratable acidity, °Brix and pH, is related to the degree of maturity of the product. For instance, a product with a greater °Brix value can have a lower % of citric acid [48]. According to previous research [49], the reported °Brix of tomatoes is 4.94 and 0.35% of citric acid. The °Brix values are close to the those obtained in this research, exhibiting variation in titratable acidity. For both products, as the number of days passed, the °Brix increased. There was a greater increase in the C- samples that had no biopackaging. Furthermore, the C- samples had greater maturity, which affected their shelf life, while those with the coating had a lower content of °Brix. The results are shown in Table 7.

#### 3.10.2. Weight Loss (WL)

Figure 8 shows the weight loss of the tomato fruits. In comparison with day 1, on day 9, there was a loss of tomato weight due to the dehydration process due to respiration, which is considered the main cause of the decrease in fruit and vegetable fruits and vegetables. This water loss is related to the pressure difference between the surrounding atmosphere and the surface of the fruit [50]. A greater loss percentage was observed for the C+ group, with a percentage of more than 7%. On the other hand, for the C- group, the percentage was less than 4% and for the samples with packaging using betalains, the percentage of loss was less than 3%, which is the lowest percentage of loss. According to [51], the use of biodegradable bags resulted in a loss of 2.5%, which is similar to the results obtained for the sample with coating.

#### 3.10.3. Shelf Life

The determination of the shelf life of the tomato fruits without (C-) and with (1C+) biopackage showed growth of *B. cinerea*, and 2C+ increased only its maturity (Figure 9). However, it inhibited the growth of the fungus, while the sample containing microcapsules with betalains inhibited the fungus, resulting in a lower maturity index and less dehydration.

Based on [52], the shelf life of ball tomatoes is between 10 and 12 days, if a temperature of 10 °C is applied. The shelf life of coated tomatoes increased by up to 18 days when they were stored at a temperature of 4–5 °C. According to [53,54], the delayed deterioration of the fruits could be attributed to the antimicrobial activity of the tested films, showing inhibition of *B. cinerea* by the packages. The release of betalains occurs from day 8, which is the common shelf life of this tomato, which can help to protect tomato from the growth of this fungus. Thus, the biopackages are a good alternative for extending the shelf life of tomatoes (see Figure 8).

## 4. Conclusions

The influence of microencapsulated betalains based on garambullo in the preparation of chitosan–PVOH films was investigated. In addition, the application of these biopackages for tomato coating was reported. The behavior of the biopackages was characterized using physical tests, water vapor permeability, puncture tests, extension, color, differential scanning calorimetry (DCS), and Fourier transform infrared (FTIR) spectroscopy. Also, the antioxidant and antimicrobial activities of the biopackages were studied. The proposed biopackages exhibited good water vapor permeability characteristics and good mechanical properties. The biopackages registered a greater content of total phenols in comparison with the controls due to the presence of betalains that registered phenolic compounds, allowing the in vitro inhibition of *B. cinerea.*

The application of the proposed coating on the tomatoes increased their shelf life by 80%. Thus, the proposed biopackages have potential application in the coating of fruits and vegetables.

## Figures and Tables

**Figure 1 antioxidants-13-01031-f001:**
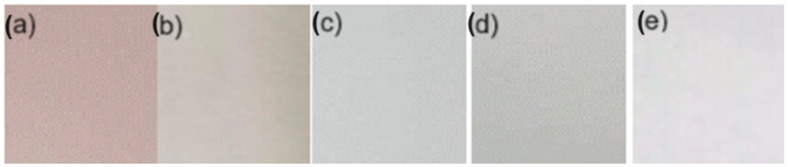
Biopackaging: (**a**) sample (S) PVOH-CS-MBT based, (**b**) positive control (C+) PVOH-CS-MBL based, (**c**) negative control 1 (C-1) PVOH-CS based, (**d**) negative control 2 (C-2) CS based, and (**e**) negative control 3 (C-3) PVOH based.

**Figure 2 antioxidants-13-01031-f002:**
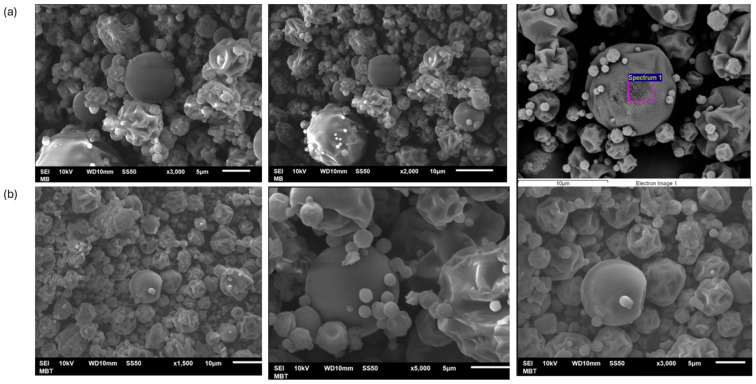
SEM images of microcapsules. (**a**) control microcapsules and (**b**) Microcapsules with juice of garambullo (*Myrtillocactus geometrizans*).

**Figure 3 antioxidants-13-01031-f003:**
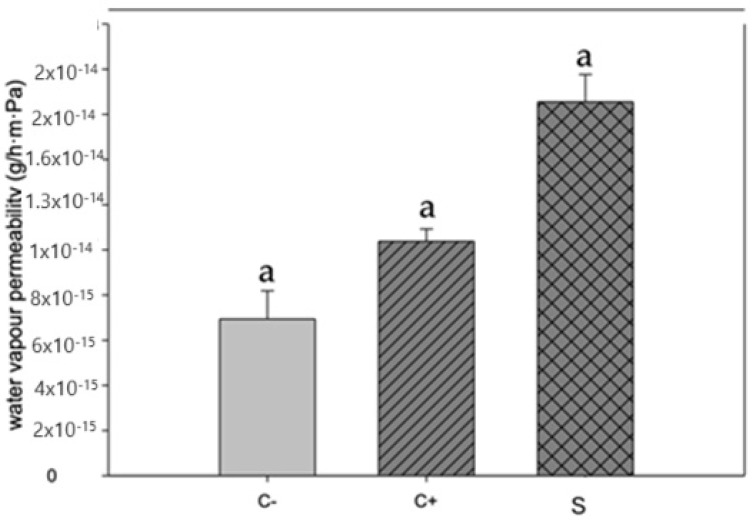
Water vapor permeability in biopackage based on CS-PVOH, control microcapsules, and microcapsules with garambullo juice. Letter (a) within each column indicate significant differences for each variable (*p* = 0.05).

**Figure 4 antioxidants-13-01031-f004:**
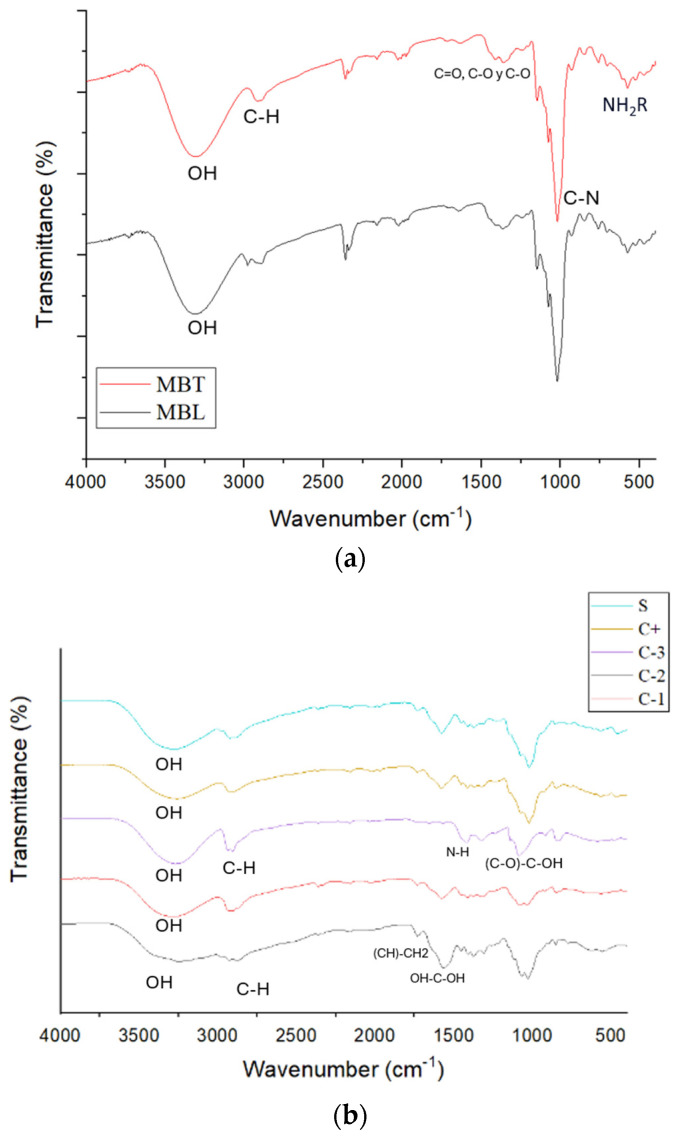
FTIR spectrum of (**a**) MBL and MBT microcapsules and (**b**) C-1, C-2, C-3, C+, and S.

**Figure 5 antioxidants-13-01031-f005:**
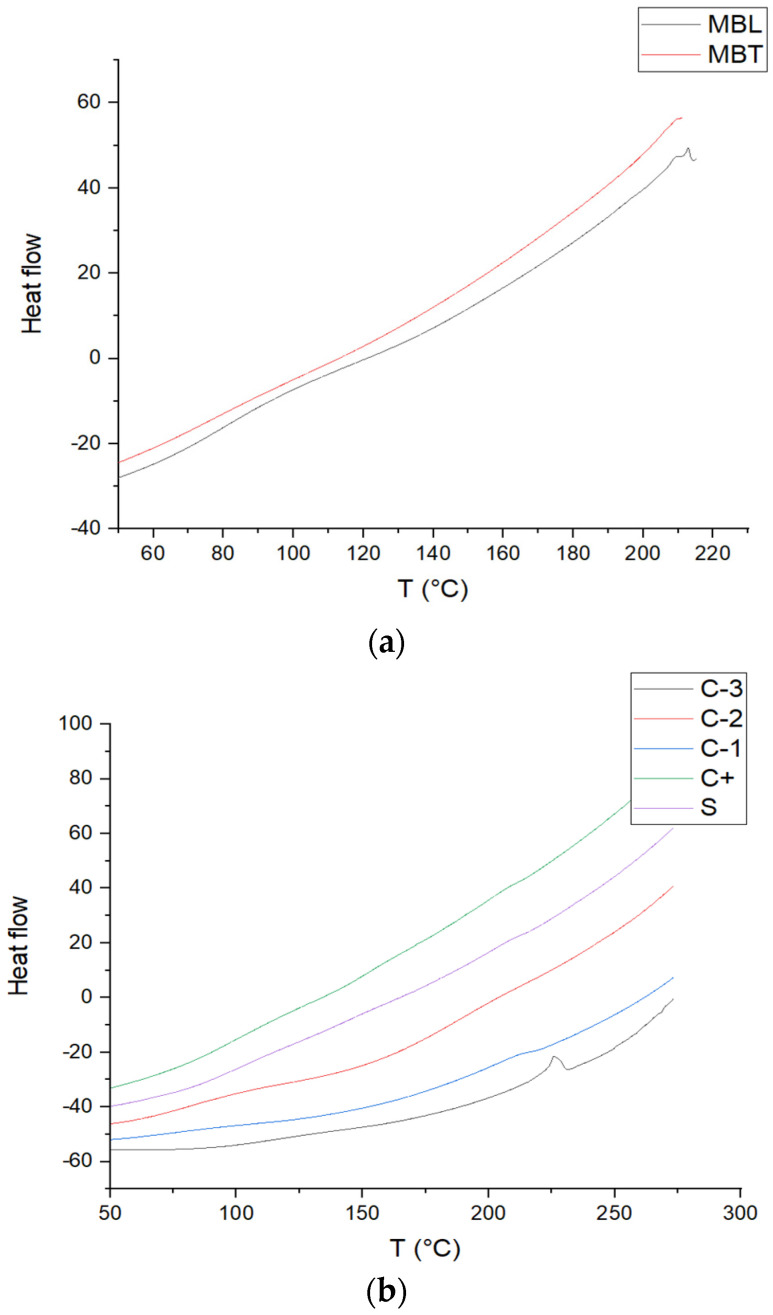
DSC curves of (**a**) MBL and MBT microcapsules, and (**b**) C-1, C-2, C-3, C+, and S.

**Figure 6 antioxidants-13-01031-f006:**
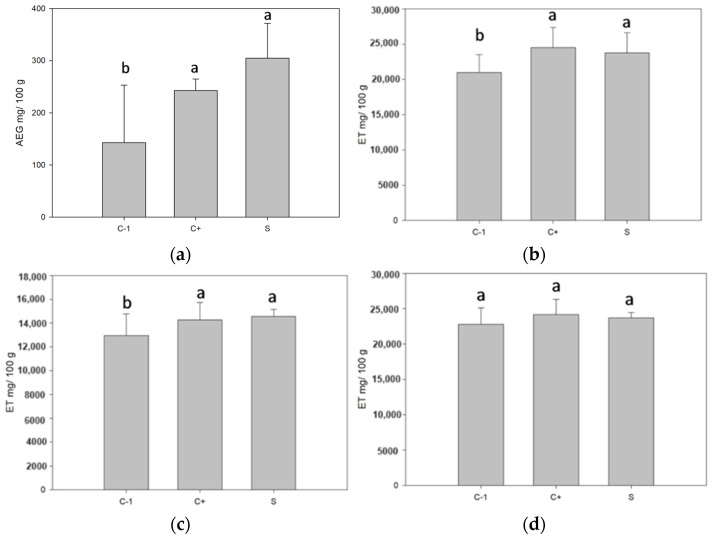
Phenolic content and antioxidant activity of biopackages from CS-PVOH, control microcapsules, and microcapsules with garambullo juice: (**a**) total phenols, (**b**) DPPH determination, (**c**) ABTS determination, and (**d**) FRAP determination. Different letters (a, b) within each column indicate significant differences for each variable (*p* = 0.05).

**Figure 7 antioxidants-13-01031-f007:**
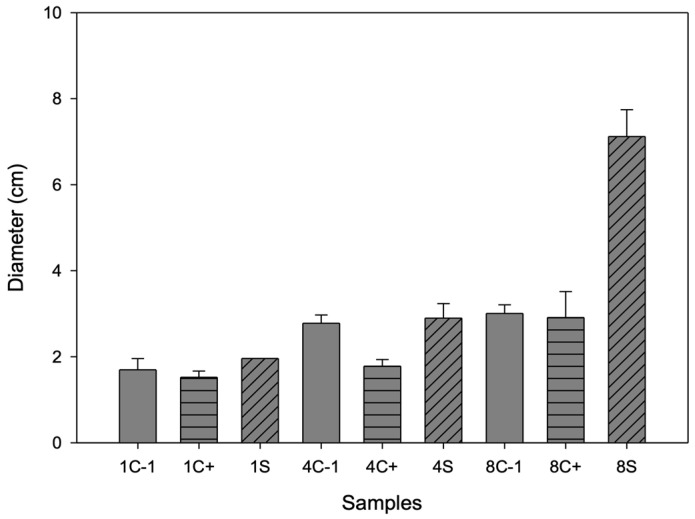
Inhibition halo diameter of biopackages from CS-PVOH, control microcapsules, and microcapsules with garambullo juice against Botrytis cinerea on days 1, 4, and 8.

**Figure 8 antioxidants-13-01031-f008:**
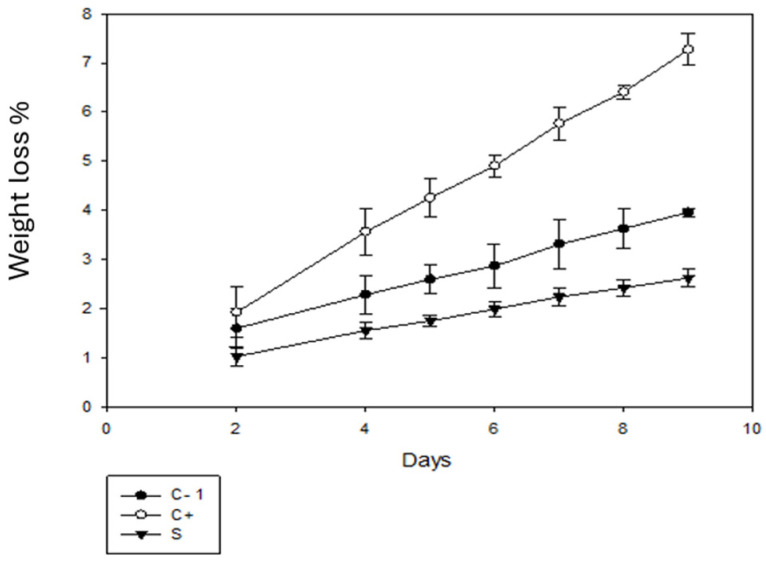
Percentage weight loss of tomato without packaging (C-), and with C+ and S packaging.

**Figure 9 antioxidants-13-01031-f009:**
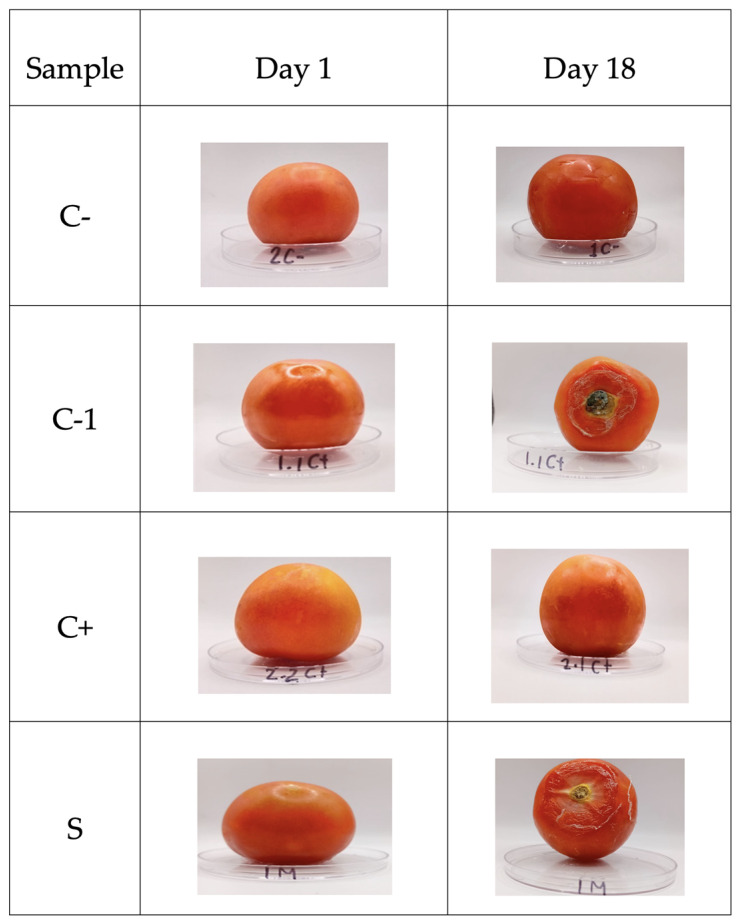
Shelf-life of tomatoes. C- refers to samples without biopackaging, C-1 refers to samples with bi-opackaging, C+ refers to samples with biopackaging and control microcapsules, and S refers to biopackaging samples with microcapsules of betalains.

**Table 1 antioxidants-13-01031-t001:** Color characterization of microcapsules with garambullo juice.

T (°C)	L	a	b	C	h
160	87.1 ± 2.4	13.7 ± 2.1	1.2 ± 0.1	13.8 ± 2.1	355.0 ± 0.5

**Table 2 antioxidants-13-01031-t002:** Betalain content, betacyanin content, betaxanthin content, water activity, and condensed tannins in microcapsules with garambullo juice.

Betacyanin (mg Pigment/100 g Sample)	Betaxanthin (mg Pigment/100 g Sample)	Total Betalain (mg Pigment/100 g Sample)	Water Activity	Condensed Tannins (Eq. of (+) Catechin mg/100 g of Sample)
6.1 × 10^−3^ ± 1 × 10^−3^	6.3 × 10^−3^ ± 1 × 10^−3^	1.22 × 10^−2^ ± 1 × 10^−3^	0.31 ± 0.005	1 × 10^−2^ ± 0

**Table 3 antioxidants-13-01031-t003:** Antioxidant characterization of microcapsules.

Phenols (AEG mg/100 g of Sample)	DPPH (ET mg/100 g of Sample)	ABTS (ET mg/100 g of Sample)	FRAP (ET mg/100 g of Sample)
4475 ± 907	85,971.4 ± 2439.8	21,900 ± 2509.9	133,469 ± 11,006

**Table 4 antioxidants-13-01031-t004:** Physical parameters of biopackages based on CS-PVOH, control microcapsules, and microcapsules with garambullo juice.

B	Weight (g)	Thickness (m)	Sample Area (m^2^)	Surface Density (kg/m^2^)	Density (kg/m^3^)
C+	0.2 ± 2.9 × 10^−3 a^	8.3 × 10^−4^ ± 7.6 × 10^−5 b^	4.4 × 10^−3^ ± 5 × 10^−4 a^	5 × 10^−2^ ± 9 × 10^−3 a^	6.1 × 10^2^ ± 0.92 × 10^2 a^
C-1	0.2 ± 2.9 × 10^−2 a^	8.9 × 10^−4^ ± 5.8 × 10^−5 b^	4.5 × 10^−3^ ± 5 × 10^−4 a^	5.2 × 10^−2^ ± 8 × 10^−3 a^	5.9 × 10^2^ ± 1.20 × 10^2 a^
S	0.2 × 10^−2^ ± 1.2 × 10^−3 b^	1.3 × 10^−3^ ± 2.6 × 10^−5 a^	4.6 × 10^−3^ ± 7 × 10^−4 a^	3.4 × 10^−2^ ± 3.5 × 10^−2 b^	2.7 × 10^2^ ± 0.77 × 10^2 b^

B is biopackaging, S is sample, C-1 is negative control 1, and C+ is positive control 1. Different letters (a, b) within each column indicate significant differences for each variable (*p* = 0.05).

**Table 5 antioxidants-13-01031-t005:** Mechanical properties of biopackaging based on CS-PVOH, control microcapsules, and microcapsules with garambullo juice.

B	Young’s Modulus (MPa)	Tensile Strength (MPa)	Puncture Strength (Kg)	Deformation at the Breaking Point	% Elongation
C-1	6.400 ± 2 ^a^	22.200 ± 0.4 ^a^	6.870 ± 1.5 × 10^−4 c^	0.569 ± 2 × 10^−3 b^	5.700 ± 0 ^a^
C+	6.200 ± 1.4 ^a^	23.500 ± 2.5 ^a^	9.810 ± 1.2 × 10^−4 b^	1.620 ± 3 × 10^−3 a^	4.500 ± 0 ^c^
S	5.500 ± 8 ^a^	22.800 ± 3 ^a^	1.262 × 10^−3^ ± 0 ^a^	1.620 ± 8 × 10^−4 a^	5 ± 0.5 ^b^

B is biopackaging, S is sample, C-1 is negative control 1, and C+ is positive control 1. Different letters (a, b, c) within each column indicate significant differences for each variable (*p* = 0.05).

**Table 6 antioxidants-13-01031-t006:** Color parameters in biopackages based on CS-PVOH films, control microcapsules, and microcapsules with garambullo juice.

	Color				
B	L	a	b	C	h
C-1	83.0 ± 0.2 ^a^	4.0 ± 0.02 ^b^	−0.4 ± 0.5 ^b^	4 ± 0 ^b^	19.4 ± 1.4 ^a^
C+	81.9 ± 0.3 ^b^	4.2 ± 0.20 ^a^	1.1 ± 0.6 ^a^	4.9 ± 0.1 ^a^	13.8 ± 6.8 ^a^
S	81.7 ± 0.5 ^b^	4.6 ± 0.01 ^a^	1.5 ± 0.3 ^a^	4.8 ± 0.1 ^a^	17.9 ± 3.4 ^a^

B is biopackaging, S is sample, C-1 is negative control 1, and C+ is positive control 1. Different letters (a, b) within each column indicate significant differences for each variable (*p* = 0.05).

**Table 7 antioxidants-13-01031-t007:** Physicochemical properties of tomatoes on days 1 and 9 after coating with CS-PVOH, control microcapsules, and microcapsules.

Physicochemical Properties	Day 1 S	Day 9 S	Day 9 C+	Day 9 C-1
pH	4.5 ± 0.0 ^b^	4.6 ± 0.0 ^b^	5.1 ± 0.2 ^a^	4.6 ± 0.2 ^b^
°Brix	4.5 ± 0 ^b^	5.5 ± 0.3 ^a^	4.6 ± 0.1 ^b^	5.5 ± 0.2 ^a^
Titratable acidity g acid/100 mL sample	1.2 ± 0.2 ^a^	1.0 ± 0.0 ^b^	0.9 ± 0.1 ^b^	0.8 ± 0.0 ^b^

Different letters (a, b) within each column indicate significant differences for each variable (*p* = 0.05).

## Data Availability

Data are contained within the article.

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
