# Peer review of "Microencapsulation of Betalains Extracted from Garambullo (Myrtillocactus geometrizans) to Produce Active Chitosan–Polyvinyl Alcohol Films with Delayed Release of Bioactive Compounds"

_antioxidants, 2024, doi:10.3390/antiox13091031_

Round 1

Reviewer 1 Report

The research is original but poorly presented. Major revisions must be done before further evaluation.

The research is original but poorly presented. Major revisions must be done before further evaluation. Please see the attached file

Author Response

Agregamos las redes de comunicación en el archivo adjunto. Por favor, vea el archivo adjunto.

Reviewer 2 Report

The chapter "Materials and methods" should be completed with some observations.

Please see in the attach.

Author Response

we add the commnets in the attached file. Please see the attachment.

Reviewer 3 Report

The manuscript “Microencapsulation of betalains extracted from garambullo (Myrtillocactus geometrizans) to produce active Chitosan-PVA 3 films with delayed release of bioactive compounds is interesting, well performed, well conducted and well organized.

1-Lines 80-82: In this work, we propose an effective method to preserve the main components in 80 garambullo juice using a microencapsulation process. We present the influence of mi- 81 croencapsulated betalains based on garambullo in the fabrication of chitosan-PVA films 82 and their application in tomate coating.

2-Line 92: 30 % soluble solids, which were accepted in the spray drying process, composed by 11.03 % garambullo juice, 0.95 % Aloe vera mucilage, 9.48 % starch, 0.1% Tween 20, 93

3- Figure 1 is not cited in the manuscript.

4- Moderate editing of English language is  required. 

The manuscript “Microencapsulation of betalains extracted from garambullo (Myrtillocactus geometrizans) to produce active Chitosan-PVA 3 films with delayed release of bioactive compounds is interesting, well performed, well conducted and well organized.

1-Lines 80-82: In this work, we propose an effective method to preserve the main components in 80 garambullo juice using a microencapsulation process. We present the influence of mi- 81 croencapsulated betalains based on garambullo in the fabrication of chitosan-PVA films 82 and their application in tomate coating.

2-Line 92: 30 % soluble solids, which were accepted in the spray drying process, composed by 11.03 % garambullo juice, 0.95 % Aloe vera mucilage, 9.48 % starch, 0.1% Tween 20, 93

3- Figure 1 is not cited in the manuscript.

Author Response

We add the commnets in the attached file. Please see the attachment.

Round 2

Reviewer 1 Report

All the requested revisions have been made by the authors.

The manuscript is optimized. I have no additional comments. I can suggest accepting it in its current form.

Best regards!

All the requested revisions have been made by the authors.

The manuscript is optimized. I have no additional comments. I can suggest accepting it in its current form.

Best regards!